# Folding and Intrinsic Disorder of the Receptor Tyrosine Kinase KIT Insert Domain Seen by Conventional Molecular Dynamics Simulations

**DOI:** 10.3390/ijms22147375

**Published:** 2021-07-09

**Authors:** Julie Ledoux, Alain Trouvé, Luba Tchertanov

**Affiliations:** Centre Borelli, CNRS, ENS Paris-Saclay, Université Paris-Saclay, 4 Avenue des Sciences, F-91190 Gif-sur-Yvette, France; julie.ledoux@ens-paris-saclay.fr (J.L.); alain.trouve@ens-paris-saclay.fr (A.T.)

**Keywords:** receptor tyrosine kinase, RTK, KIT cytoplasmic region, kinase insert domain, KID, molecular dynamics and folding, conformational plasticity, intrinsically disordered region, transient structures, free energy landscape

## Abstract

The kinase insert domain (KID) of RTK KIT is the key recruitment region for downstream signalling proteins. KID, studied by molecular dynamics simulations as a cleaved polypeptide and as a native domain fused to KIT, showed intrinsic disorder represented by a set of heterogeneous conformations. The accurate atomistic models showed that the helical fold of KID is mainly sequence dependent. However, the reduced fold of the native KID suggests that its folding is allosterically controlled by the kinase domain. The tertiary structure of KID represents a compact array of highly variable α- and 3_10_-helices linked by flexible loops playing a principal role in the conformational diversity. The helically folded KID retains a collapsed globule-like shape due to non-covalent interactions associated in a ternary hydrophobic core. The free energy landscapes constructed from first principles—the size, the measure of the average distance between the conformations, the amount of helices and the solvent-accessible surface area—describe the KID disorder through a collection of minima (wells), providing a direct evaluation of conformational ensembles. We found that the cleaved KID simulated with restricted N- and C-ends better reproduces the native KID than the isolated polypeptide. We suggest that a cyclic, generic KID would be best suited for future studies of KID f post-transduction effects.

## 1. Introduction

Receptor tyrosine kinases (RTKs) act as sensors for extracellular ligands, the binding of which trigger dimerisation of RT, activation of its kinase function and auto-phosphorylation of specific tyrosine residues in the cytoplasmic domain [1,2]. This leads to the recruitment and activation of multiple downstream signalling proteins, which carry the signal to the nucleus, where it alters patterns of gene transcription such as those governing various aspects of the cell physiology. Initiation of this cascade-like process involves different regions of the multidomain RTKs, each of them performing specific actions that are finely concerted by a tightly regulated allosteric mechanism controlling all functional processes of RTK [3]. Explicit elucidation of the signalling cascade represents an important and unsolved problem in cell biology.

The modular extracellular domain (ED) of RTK, containing characteristic structural motifs involved in specific ligand binding, is formed by various motifs (Ig-like, cysteine-rich, cadherin fragments, etc.) interconnected by coiled linkers, providing high conformational plasticity (Figure 1A). The binding of ligands affects the monomer–dimer equilibrium of RTKs by stabilising the dimeric state through a global conformational change in the ED [4]. The signal induced by ligand-binding propagates across the transmembrane (TM) domain to the cytoplasmic domain (CD) and promotes its activation. The cytoplasmic domain of RTKs also has a modular structure composed of the juxtamembrane region (JMR), the split tyrosine kinase (TK) domain with the proximal (N-) and distal (C-) lobes linked by a kinase insert domain (KID), and the C-terminal tail (C-term).

We concentrate our study on the RTK KIT from the PDGFR (III-type) family. The physiological actions of KIT controlling cell survival, proliferation, differentiation, and migration depend on the activation of specific or overlapping pathways [5], which endows the activity of the SCF/KIT system (where SCF is the stem cell factor that regulates the KIT activation and therefore, triggers the initiation of multiple signal transduction pathways) of great complexity. Aberrant regulation of KIT signalling networks is associated with the progression of many cancer types, including human acute myeloid leukaemia, aggressive systemic mastocytosis, melanoma, gastrointestinal stromal tumour, and stomach cancers [6,7,8]. Disclosure of the KIT activated pathways in carcinogenesis will be a crucial step towards the development of KIT targeted therapies [9].

Functions of the TK domain of KIT, similarly to other RTKs, are mainly attributed to catalytic activity and trans-phosphorylation, while the post-transduction processes and the binding of intracellular proteins are associated with JMR, KID and C-terminal, which are the regions possessing multiple phosphorylation sites [1] (Figure 1B). The overall structural feature of these flexible fragments is the intrinsic disorder which is a vital condition for the creation of the dynamic networks of interacting proteins [10].

**Figure 1 ijms-22-07375-f001:**
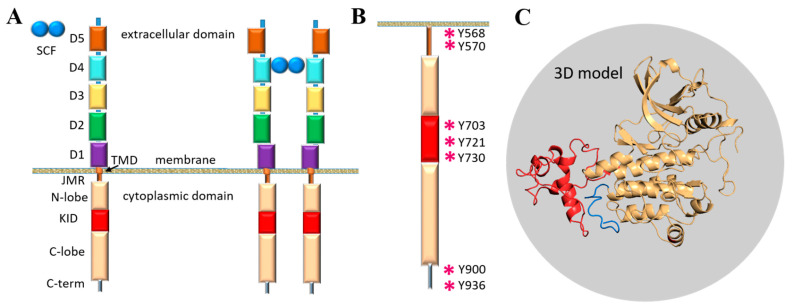
The modular structure of RTKs illustrated with KIT, a member of the RTK family III. (**A**) Structural composition of KIT: an extracellular domain (ECD) with five Ig-like regions (D1–D5), a transmembrane domain (TMD) and a cytoplasmic domain (CD) comprising a juxtamembrane region (JMR), an ATP-binding region (N-lobe), the phosphotransferase domain (C-lobe) with a kinase insert domain (KID) and a C-terminal. The stem cell factor (SCF) extracellular binding induces dimerisation and activation of KIT. (**B**) The tyrosine residues (Y, showed by asterisk) of JMR, KID and C-terminal identified as the phosphorylation sites involved in the recognition of cellular partners. (**C**) 3D structural model of the KIT CD with KID (red) and C-term (blue) [11].

A first structural model of the full-length cytoplasmic domain of KIT has been published (Figure 1C) [11]. The primary analysis of the structural and dynamical properties of this model showed that the conformational variability of KIT is provided mainly by JMR, KID and C-terminal, which were interpreted rather as intrinsically disordered (ID) regions demonstrating significant structural and conformational plasticity.

Because ID proteins (IDPs) lack stable secondary structures (2D) under physiological conditions, they exist as heterogeneous conformational (3D) ensembles, and are capable of rapidly changing conformation upon influence by an effector (e.g., binding of ligand/cofactor/protein). Principally, the determination of a single structure of the ID regions or ID proteins has no physical relevance, as such structures present only an isolated element (a lone conformation) from a huge conformational space. A more pertinent approach is the description of such proteins in terms of the probabilistic population of the different regions, and the correlation of these probabilities with the protein function. Most experimental techniques employed to study IDPs [12] suffer from a conundrum: the empirical observables, which make it possible to assess the protein disorder, represent an average over the conformational ensemble, but the ensemble itself cannot be unequivocally inferred from the experiments. Therefore, computational methods provide an advantageous approach for analysing IDPs. In particular, molecular dynamics (MD) simulation provides great insights into this challenge [13], as an approach that can, in principle, produce the structural ensemble of biomolecules.

In the present study, the focus is on the KID of KIT, a binding hub assuming exquisite specificity of the receptor. As has been reported that five functional phosphorylation sites of KID from KIT, three tyrosine (Y703, Y721, Y730), and two serine (S741 and S746), provide the alternative binding sites for the adaptors, signalling and scaffolding proteins in the cytoplasm [14]. Phosphorylation of Y703 supplies the binding site for the SH2 domain of Grb2, an adaptor protein initiating the Ras/MAP kinase signalling pathway. Phosphorylated Y721 and Y730 are the recognition sites of PI3K and phospholipase C (PLCγ), respectively. The function of Y747 has not yet been described. Phosphorylated serine residues, S741 and S746, bind PKC (protein kinase C) and contribute to re-control of PKC activity under the receptor stimulation. The functional importance of KID is also emphasised by newly identified mutations of K704, N705 and S725, which were reported as activating in gastrointestinal stromal tumours [15]. Consequently, a study of KID structure-dynamics features and their relation to KID function is still crucial but/and obviously challenging.

The conformational dynamics of KID from KIT was probed by conventional molecular dynamics (cMD) simulations. This method generates the atomic-level data required for a detailed analysis of the conformational space and the identification of folding intermediates related to function and/or characterisation of functionally important phenomena related to allosteric regulation [16]. Allostery is often a fine adaptive mechanism of protein modulation in the cellular environment (e.g., adaptation of binding partners to form biomolecular assemblies) during signal transduction, catalysis, and gene regulation [17,18].

One approach for studying the assembly of multidomain proteins and their folding is to use the modular domain of the protein, which preserves binding capabilities even when the domain is removed from the context of the full-length protein [19]. The ability of modular protein domains to independently fold and bind both in vivo and in vitro has been taken advantage of by a significant portion of proteomics studies that have used modular domains to assess the protein–protein interactions required for a diverse set of cellular processes, including signal transduction and subcellular localisation.

The use of KID as a cleaved polypeptide represents a promising strategy for the exploration (empirically and numerically) of KIT signal transduction and the modulation of protein function by controlled interference with the underlying molecular interactions. Such use will be fully justified if we can prove that KID of KIT is a modular domain that preserves its structural and dynamic properties. As we seek draw conclusions with respect to the usefulness or the disadvantage of using cleaved KID as a reduced model to study the RTK KIT ‘interactome’ [20], a comparative analysis of this model with the KID of KIT was carried out. We also suggest a possible dependence of KID on the kinase domain of KIT, which can either be induced locally by geometric restriction on the KID terminus or globally promoted by long-range allosteric effects.

Our present study of KID from KIT focuses on its folding features and intrinsic disorder with the aim of defining a KID species suitable for the exploration of the KIT ‘interactome’.

## 2. Results

### 2.1. Data Generation

The kinase insert domain of KIT, composed of the 80 amino acids (aas) (F689–D768), was examined by conventional MD simulations (all-atom, with explicit water) as the cleaved polypeptide (KID^C^) and as the subdomain of the kinase domain (KID^D^ from KIT). The cleaved KID was simulated as a fully unconstrained entity (KID^C^), and as a polypeptide with restrained distance between two Cα-atoms of terminal residues F689 and D768 (cleaved restrained KID, KID^CR^), which is conserved in crystallographic structures of KIT [11]. The MD simulation run (of 1.8 µs for KID^C^ and 2 µs for KIT and KID^CR^) was repeated four (KID^C^) and two (KIT and KID^CR^) times with different randomised initial atomic velocities to extend conformational sampling of each studied entity and to examine the consistency and completeness of the produced KID conformations. The other parameters of the MD simulations of KID and KIT were strictly identical. Each simulation was started with an equilibrated structure obtained after minimising the neutralised solvated protein which is either a 3D de novo model of KID or of KIT.

### 2.2. General Characterisation of MD Simulation Data

The data analysis was performed on each trajectory after the least-square fitting of MD conformations to the initial KID conformation, to avoid the motion of the domain as a rigid body.

First, the global stability of KID throughout the MD simulations was estimated using the root mean square deviations (RMSDs) computed on the Cα atoms relative to the initial KID model (at t = 0 µs) as a frame of reference. For the cleaved KID, simulated as the unconstrained polypeptide (KID^C^), the RMSD profiles differ among trajectories 1–4, and their values vary along each trajectory, but the ranges of RMSD variations are comparable between trajectories (Figure 2A, top panel). Well-resolved slopes on RMSD curves are either a single event showing a rapid increase of RMSD values, or a two-event process showing an alternating increase/decrease in RMSD. Such sudden changes in RMSD, observed at t = 0.40–0.55, 0.77–1.79 and 1.55–1.60 µs in trajectories 1, 2 and 4, respectively, reflect regular conformational transitions in KID suggested in [11] based on a single simulation. The RMSDs cover a large range of values (4–9 Å) arranged in two main peaks per trajectory with different (most trajectories) or comparable (one trajectory) populations (Figure 2B, top panel). Two main peaks in each replica are separated by 1–3 Å, while the most populated peaks, composed of similar conformations, from replicas 1–4 are shifted only by ~1 Å. While RMSD of MD conformations were calculated from the initial coordinates, which are identical in all trajectories, we can suggest (1) at least two groups of highly different KID conformations within each trajectory; (2) partial overlapping of conformations from different trajectories.

Similar to the cleaved KID, the RMSD curves of KID^D^ show either a rather dramatic increase or a series of small reversal changes (Figure 2A, mid-panel). The probability distribution of such RMSD variations appears as two distinct well-separated peaks or two overlapped distributions in a slightly narrow range with respect to KID^C^ (Figure 2B, mid-panel). RMSD of the cleaved KID simulated with the restrained distance between its N- and C-end vary within a range similar to KID^D^; nevertheless, the probability distribution shows a single peak for each replica (Figure 2A,B, low panel).

Second, the root mean square fluctuations (RMSFs) were calculated for each residue of KID. As expected, the highest RMSFs are mainly observed for residues at KID^C^ extremities, while the fluctuations of these residues in KID^D^ and KID^CR^ are very limited. Examination of inter-residual contacts in KID^D^ led to the identification of two stable (occurrence of 97–99%) hydrogen bonds N–H⋯O liking two pairs of residues, C691–L764, and L766–F689, which maintain the N- and C-end of KID in very proximal position (Figure 2, right; Appendix A). Apparently, these strong H-bonds restrain the mobility of KID extremities as reflected in small RMSFs. Similar to KID^D^, the N- and C-ends of cleaved KID are stabilised by H-bonds, but the other pairs of residues are involved.

Curiously, in all KID entities three hydrophobic residues, V731, V732 and P733, positioned on the random coil show small RMSFs in respect to preceded and followed residues, therefore, the RMSF curves in the different entities of KID display a comparable profile, which is described as the ‘camel double-humped’ contour (KID^C^) or close to this contour (KID^D^ and KID^CR^) (Figure 2C).

The two descriptors, RMSD and RMSF, indicate the highly heterogeneous conformational composition of each data set obtained by cMD simulation of each KID entity. This data reveals the intrinsically disordered nature of KID from KIT, previously suggested in [11] based on a single trajectory of MD simulation. It is well known that a high content of polar and charged residues increases the propensity of a protein to be disordered [21,22,23]. The KID of KIT, composed of 58% such residues, is a good candidate for being an intrinsically disordered region, and this feature is probably only sequence-dependent and disconnected from the context of KID as an entity sampled by MD simulation, either as a cleaved isolated polypeptide or a domain of KIT.

Heterogeneous KID conformations were analysed with the ensemble-based clustering using the RMSD criterion [24]. To better grasp the conformational diversity of each studied KID entity, the conformations were clustered with the RMSD threshold r = 3, 4 and 5 Å. With *r* of 4 Å, we obtained a reasonably limited number of clusters regrouping almost all the conformations generated on each trajectory (Appendix A). Since some clusters’ representative conformations of different replicas showed striking similarity, we suggest that the conformational spaces generated by the independent trajectories of each KID entity partially overlap. Clustering analysis of the merged (concatenated) trajectory with the same RMSD threshold (r = 4 Å) produced the number of clusters which is less than a sum of clusters obtained for each replica. Such results do not contradict the supposed overlap of conformational spaces sampled by replicated trajectories.

### 2.3. Folding and Compactness of KID

The secondary structure interpretation (DSSP) indicates that the helical fold of KID is constituted of α- and 3_10_-helices (Figure 2C; Appendix A). These helices are varied in number, length and α/3_10_ ratio. These variations are observed within a trajectory, between the trajectories, and for different KID species, but the position of some helices is curiously conserved.

KID^C^ is composed of six helices, H1–H6i is made up of 44–56% of amino acids. H1-helix, the largest (15–16 aas) of all KID helices, is a long-leaved α-helix well-conserved in all replicas. Other helices of varying lengths are rather transient, switching between α- and 3_10_-helices (H2, H3 and H5) or between 3_10_-helix and random coil, which is partially folded as reverse turn or bend (H4 and H6). Similarly, KID^D^ shows a helical fold, but the number of helices (H1, H2 and H3 or H5 on average structure) and the portion of residues (25–30%) constituting these helices are significantly diminished. In KID^D^, like KID^C^, H1-helix is the largest compared to other helices; however, its length is reduced to 11–12 aas. The H1-helix is long-lived, while the other helices are fully transient.

The highly diminished folding in KID^D^ prompts two hypotheses: 

**Hypothesis** **1.***The KID folding depends on the KID ends, which are stabilised by strong H-bonds in KID fused to KIT and highly flexible in the cleaved KID*.

**Hypothesis** **2.***The KID folding depends on the status of KID as an entity, which is either autonomous polypeptide or collateral subdomain influenced by the kinase domain*.

The cleaved KID, simulated with the constrained distance between the terminal residues (KID^CR^) showed that 30–35% of amino acids are involved in regular structures, the extended and long-lived α-helix H1 (15–17 aas) and three to five transient helices, indicating that the KID^CR^ folding (at the 2D level) is more ordered than in KID^D^ and less than in KID^C^ (Appendix A). It seems that the constrained polypeptide in general better represents the folding of the native KID than the unconstrained, but the expected equivalence was not observed. The mid-domain of KID is organised similarly in KID^CR^ and KID^D^, while folding of C-terminal residues is quasi-identical in KID^CR^ and KID^C^. It is probable that the applied constraints in KID^CR^ were rather soft compared to the restricted intra-protein geometry in KID^D^. The increased flexibility of border residues in KID^CR^ compared to KID^D^, as seen by RMSFs, and the greater similarity of 2D folding of residues from C-end (H6) in KID^CR^ and KID^D^, support this hypothesis. On the other hand, we can suggest a certain bias of such comparison derived from a different number of independent MD trajectories that is 4 for KID^C^ and 2 for KID^D^ and KID^CR^. Interestingly, in all studied KID entities, residues forming H1 and H5 helices show the smallest RMSF values compared to the other helices.

The three-dimensional (3D) structure of KID in all studied entities represents a compact array of αH- and 3_10_-helices linked by short or extended loops (random coils or turns) that play a principal role in the conformational diversity of KID. In the most of conformations, KID retains its collapsed globule-like shape that is slightly elongated towards the KID connected to the kinase domain, and therefore, is best described as a flattened (oblate) ellipsoid with an opening given by the distance between its ends (Figure 2). This opening in KID^D^ is controlled by hydrogen bonds linking residues from N- and C-extremities. The enlarged displacement of the border residues in the cleaved KID promotes a diminishing or a full destabilisation of such H-bonding.

To characterise the size and compactness of KID, the mass-weighted radius of gyration (Rg) was calculated for each entity of KID on MD conformations. We compared the distribution of Rg vs. RMSD between each ensemble of KID conformations (on every trajectory on each KID entity) (Appendix A). All distributions show at least two heavily populated regions separated by an area with a lower number of intermediates. The mean values of Rg for each peak are comparable in the two replicas on KID^D^ (and on KID^CR^), but are slightly different between entities. In KID^D^, the Rg of each peak (mv of 11.80(1) and 12.22(1) Å) shows a slightly increased value in comparison with KID^CR^ (mv of 11.33(1) and 12.00(2) Å). In KID^C^, the Rg mean values of the heavily populated regions vary between replicas, but these variations spanned (cover) the Rg range observed in KID^D^ and KID^CR^ (mv of 11.64(1) and 12.52(2) Å).

Since strong variations in Rg (and RMSD) are observed along the same trajectory on cleaved KID, this indicates a higher conformational variability of KID^C^, rather than more exhaustive sampling over the four replicates. Additionally, since the cluster analysis of the individual and concatenated trajectories showed at least a partial overlap of the conformational spaces generated on the replicas for each KID entity (Appendix A), the more in-depth analysis of each KID entity was performed on the concatenated data.

### 2.4. Folding and Intra-Domain Interactions of KID

The propensity of globular proteins to be compact is the key reason that their folded states achieve high packing density. We suggest that the contact map can characterise KID collapsibility. The high instability of folded structures, which are, except for H1, transient, converting between αH- and 3_10_-helices or between helix and random coil, together with their great mobility, producing an expected irregularity of contacts, resulted in the smeared pattern on the contacts maps (Figure 3A).

The contact maps are sufficiently different between replicas of the same entity of KID (e.g., replicas 1–4 of KID^C^), but strikingly similar between distinct KID entities (e.g., replicas 3 on KID^C^ and 1 on KID^CR^, or replicas 4 on KID^C^ and 2 on KID^CR^). Such patterns in the contact map can be linked, on the one hand, to a large difference in folding of the same KID entity observed in the MD replicas, and on the other hand, to a partial similarity of the conformational and structural characteristics of the different entities of KID.

According to the contact maps, the residues of N- and C-ends of the cleaved KID, despite their high (in KID^C^) or moderate (in KID^CR^) flexibility, are still involved in intramolecular contacts. Apparently, the change in conformation in the cleaved KID leads to the appearance of alternative hydrogen bonds, which are formed by the same residues as in KID^D^, and the other neighbour residues (C691 ⋯ D765, S692 ⋯ D765, L766 ⋯ F689, E767 ⋯ I690).

The high-occurrence of contacts between multiple residues from the folded and randomly coiled segments of the mid-domain of KID^C^ indicate diverse non-covalent interactions between helices (H1, H2 and H5), helices and coiled regions, and randomly coiled regions. This large number of intra-molecular contacts suggests that multiple regions contribute to maintaining the inherent (intrinsic) 3D structure of KID in all studied entities, independently of the fold.

While a considerable portion of KID undergoes both large conformational changes and high fluctuations, we focused on residues showing low fluctuations (RMSF)—the ‘pseudo-rigid’ segments E699-S709, P754-L764 and V731-P733—and analysed their contacts with all the other residues (Figure 3B,C). It is interesting to note that these ‘pseudo-rigid’ fragments show a different fold, a long-leaved helix (H1), a random coil linking H3 and H4 helices, and a transient helix (H5), respectively.

The most fascinating observations from the analysis of contacts formed by the ‘pseudo-rigid’ fragments are as follows: (i) a large number of KID residues are involved in intramolecular contacts, (ii) the charged and polar ‘contacting’ residues are the most abundant compared to hydrophobic residues, (iii) residues V728–D737 constituting the linker connecting H3 to H4 or H5 are regularly involved in intramolecular contacts, and (iv) residues of the N- and C-ends interact with the ‘pseudo-rigid’ segments. The quantitative estimation of the number of contacts stabilising KID is minimal in KID^D^, while in the cleaved KID their number is increased to 46% (KID^C^) and 52% (KID^CR^). Interestingly, there are a fairly limited number of residues that do not contribute to intramolecular interactions in KIT.

Analysis of the contribution of ‘contacting’ residues in the intramolecular H-bonds (including the salt bridges) and the hydrophobic interactions, inspected separately, showed that both types of interactions form an extended and dense network of contacts that stabilises a compact globular shape of all studied KID entities (Figure 4A). To illustrate that these complex intramolecular contacts held a ‘globule-like’ shape of KID, we depicted the H-bonds and hydrophobic interactions on a randomly chosen conformation of each entity (Figure 4B).

In particular, the helices H1 and H5 interact mainly through the multiple hydrophobic interactions stabilising their proximal position. Residues from H1 and H5 form H-bonds and hydrophobic interactions with residues from the loop connecting H3 and H4, and/or from H4. These abundant interactions stabilise a closed location of the structurally disordered regions of KID mid-domain to helices H1 and H5. H1 interacts with H2 and H3 mainly via the H-bonds formed by charged and polar residues. Finally, the N- and C-end residues interact with each other and with H1-helix. The charged (E758, E595, E699, E711 and E761) and polar (N705, N719) residues form the H-bonds by the interaction of their side-chains with the main-chain atoms of the other residues.

The observed patterns of non-covalent intramolecular contacts in KID display a key role of the αH1-helix which, like a drop of glue, attaches all the structural fragments of KID around them. The results obtained indicate that intramolecular interactions, van der Waals and electrostatic forces, are a dominant factor in the stabilisation of the compact ‘globule-shaped’ (collapsed) KID and that the collapse induced by the intramolecular force conquers the solvent-induced expansion.

Since the non-covalent contacts of KID are folding-dependent, we further focused on the time-connected dynamics of the KID secondary structure. To more finely compare the folding dynamics in KID entities, we carried out a study based on per-residue modelling by finite-state Markov models, which was carried out on data encoding the secondary structure for each KID residue of each replica in an estimated transition probability matrix from one folding type to another. The eight-category classification (eight-letter code) of secondary structures was used. We thus obtain for each replica a sequence of transition matrices (one per residue). A suitable distance (Fisher-Rao distance) between these families is then calculated for all the pairs of replicas. We obtain the Fisher-Rao matrix (8 × 8 of size), where the first four replicas are the cleaved KID^C^ (group C), the next two the KID^D^ (group D), and the last two the cleaved restricted KID^CR^ (group CR) (Figure 3F). Multi-Dimensional Scaling (MDS) was performed to get an ‘as isometric as possible’ embedding of the data in 2D (i.e., a representation by placing points on a plane while preserving the calculated inter-distances as well as possible). Interestingly, the three groups do not form separate clusters, as shown in the 2D representation. Group C occupies the space quite well with replica number 4 at the central position, which is also central for all KID trajectories, while the two points of group D occupy extreme positions. These observations corroborate fairly well with the secondary structures (Figure 5C,E) and the inter-distance matrix (Figure 3A).

### 2.5. Geometry of the Tyrosine Residues in KID

As KID contains four tyrosine residues, three of which are known to be phosphorylation sites (Y703, Y721, and Y730) and one (Y747) that has an unclear functional role [25], we focused on their structural features. These residues do not belong to the helices H1–H6 identified in KID, except Y703, which is positioned on a highly conserved αH1-helix. The other phosphotyrosine residues are located on transient fragments converting between α-, 3_10_-helix and random coil. This observation suggests that the higher solvent accessibility of the phosphorylation sites afforded by the absence of secondary structure facilitates the post-transduction (translational) modifications required for recognition and recruitment of downstream molecules that are adaptors or signalling proteins.

We suggest that the geometry of the phosphotyrosine residues, key elements for substrate binding, may reflect (recover) the structural or conformational features of KID. Geometry was described by using a tetrahedron designed on the Cα-atoms of tyrosine residues regarded as nodes connected by edges (Figure 5). In the studied KID entities, the tetrahedron geometry varies greatly during the MD trajectories. Even if the edges of the tetrahedron are maintained over an extended period (30–50 ns), their lengths are further altered, either for all edges (synchronous transform) or only for certain edges (asynchronous transform).

Changes of the tetrahedron geometry during an MD trajectory are viewed either as instantaneous events or as stepwise processes. The long-time preservation of inter-tyrosine distances is apparently associated with the maintenance of the secondary structure in KID, while the synchronous/asynchronous change reflects the important transformation at the level of the helices. Consequently, the inter-tyrosine geometry is closely related to the folding–unfolding process in KID. Nevertheless, the other structural factors, such as relative orientation of the helices and flexibility of the coiled linkers, can contribute significantly to the high variability of tyrosine residue geometry. A coherent consequence of such variability of geometry of the tyrosine residues is the great dispersion of the hydroxyl groups (Figure 5F).

Furthermore, we focus on finding geometrically conserved elements of KID and their spatial relationships with incongruent structures. We suggest that residues Y703 (αH1-helix), V732 (H5 helix) and I756 (linker connecting H4 and H5 helices), with the smallest RMSF values, are a ‘rigid subset’ if the inter-residue distance is conserved in MD conformations. As expected, the inter-residue distances display essentially small variations compared to the distances between the tyrosine residues (Appendix A). Nevertheless, the almost invariant geometry of the most ‘rigid’ residues over the large periods of simulation time (≥1 µs) is followed either by an instantaneous change or by staggering alternations of their values. There is no obvious correlation between the geometry of a ‘rigid subset’ formed by Y703, V732 and I756, and the geometry of a tetrahedron designed on the tyrosine residues, Y703, Y721, Y730 and Y747; nonetheless, their changes without a doubt are connected.

### 2.6. Free Energy Landscape as a Quantitative Measure of Folding and Disorder in KID

A promising strategy for the in-depth analysis of a protein conformational space is to consider the ‘free energy landscape’ along specifically chosen coordinates called ‘reaction coordinates’ or ‘collective variables’, which describe the conformation of a protein [26,27,28]. Such interpretation leads to quantitatively significant results that allow comparison between different states of a protein. The relative Gibbs free energy ΔG between two or more states is a measure of the probability of finding the system in those states. Such representations of protein sampling with use of reaction coordinates can be the quintessential model system for barrier crossing events in proteins [29]. These can be estimated from incomplete sampling of the states, as long as it is an unbiased sampling.

The statistical quantities—radius of gyration, RMSD, helical folding, contacts, surface exposed to solvent—usually used for the description of the conformational properties of proteins, were regarded as reaction coordinates (collective variables) for the evaluation of the relative free energy (ΔG) and reconstruction of its landscape (Figure 6).

First, the probability (P) of Rg, P(Rg), and of RMSD, P(RMSD), were used as the reaction coordinates for the evaluation of the relative free energy (ΔG). The free energy landscape (FEL) as a function of RMSD and radius of gyration Rg ($FEL_{RMSD}^{R_{g}})$ for the concatenated replicas of KID is shown for each KID entity.

Each $FEL_{RMSD}^{R_{g}}$ shows a rugged landscape indicating a high conformational heterogeneity, which is best represented in the cleaved KID (Figure 6A). The heterogeneity likely builds up the entropy barrier and adds complexity to the free energy map, and thus limits spontaneous state-to-state transition when using conventional advanced sampling methods. Nevertheless, the $FEL_{RMSD}^{R_{g}}$ of each KID entity shows areas of minimum energy indicated by the red colour. The red areas represent more stability, while the reddened areas indicate transitions in the conformation of the protein followed by the thermodynamically more favourable state. The $FEL_{RMSD}^{R_{g}}$ of different entities of KID have a unique global energy minimum (in dark red), which is clearly different from the other local minima showed by higher energy values (four minima in KID^C^ and two in KID^D^ and KID^CR^). Regarding the $FEL_{RMSD}^{R_{g}}$ ranges, it becomes the narrowest in KID^D^ and the largest in KID^C^.

To characterise the KID conformations in terms of compactness, despite very small discrepancies in the values of Rg, the KID conformations were defined, based on mean values, as compact (11.2–11.6 Å), semi-compact (11.7–12.2 Å) and loose (12.3–12.8 Å). In KID^D^, the lowest energy well is constituted of KID conformations having a semi-compact structure (Rg of 11.83(1) Å) (Figure 6B, Appendix A). Two other wells are composed of loose conformations showing very close degrees of compaction (Rg of 12.2 and 12.3 Å).

In KID^CR^, the narrow well of the lowest energy includes the KID conformations with a compact structure (Rg of 11.33(1) Å). Two other wells contain compact and semi-compact conformations (Rg of 11.2 and 11.8 Å, respectively). The low energy well on the large-ranged $FEL_{RMSD}^{R_{g}}$ map of KID^C^ is composed of compact conformations (Rg of 11.80(1) Å); the other wells are composed of conformations having different compactness, with Rg varying from 11.4 Å (compact) to 12.8 Å (loose).

Second, the relative free energy ΔG of KID was evaluated while using as reaction coordinates the helical folding order parameter ($H_{fp}$), describing the fraction of helices (as calculated by DSSP), and the radius of gyration (Rg). The unimodal Gaussian distribution of $H_{fp}$ does not separate the KID conformations on isolated clusters but describes the range of variation of the helical content and delimits the most populated region (Figure 6C). The $H_{fp}$ and Rg values are apparently decorrelated. The free energy landscape, $FEL_{H_{fp}}^{R_{g}}$, reconstructed on these coordinates, shows a unique well for each KID entity, but their profile and position is differed between the studied KID (Figure 6D). This unique deep well, large in KID^C^ and narrow in KID^D^ and KID^CR^, is composed of conformations that show a different order of helical folding that is greater in cleaved KID (KID^C^ and KID^CR^) compared to the KID domain of KIT (KID^D^).

Third, for evaluation of the relative free energy (ΔG) of KID, the solvent-accessible surface area (SASA) and the radius of gyration (Rg) were used as reaction coordinates. Statistically, similarly to the helical content, the SASA is also represented by the unimodal Gaussian distribution, which is highly symmetric in KID^C^ and KID^D^, and slightly asymmetric in KID^CR^. The SASA and Rg are highly correlated metrics, showing the dependence of the solvent-accessible surface from the KID size. The free energy landscape $FEL_{SASA}^{R_{g}}$, reconstructed on these reaction coordinates, shows deep and shallow wells which are adjacent in each KID entity. In KID^C^ and KID^D^ these wells are composed of semi-compact KID conformations which are more exposed to solvent in KID^D^ and rather buried in KID^C^. In KID^CR^, the deeper well is populated by the compact and buried conformations.

The free energy landscape of KID, represented as a function of the radius of gyration Rg and of an alternate metric (RMSD, helical folding parameter, SASA) as the second reaction coordinate, groups the KID conformations in a statistically found collection of multi-minima, or super-wells. On each FEL we can distinguish the most probable states of different KID entities and compare their functionally related properties. We can also understand how these statistical properties vary in different parts of the energy landscape.

As emerges from the observations mentioned above, the representative conformations of the deepest well (the lowest energy minimum) located on each FEL do not show close similarity in their folding (secondary structure), tertiary organisation (3D structure) or the spatial position of tyrosine residues for the same KID entity (Figure 7, Appendix A). Nevertheless, in KID^D^ these randomly chosen conformations cognate well (RMSD of 2.0–3.1 Å), while in the cleaved KID, KID^C^ and KID^CR^, they are largely different (Appendix A).

The conformational sub-set of the deepest well on each FEL indicated the homogenous composition for each KID entity, as shown by similar first-principles metrics, RMSD and Rg , varying a little. It seems interesting to compare the similarity of the conformational subset from the deepest well of each KID entity with those obtained by ensemble-based clustering. The Rg mean values of the same KID entity computed on two conformational subsets are equivalent. This observation is valid for all the KID species studied. Comparing the most populated conformational subsets generated by ensemble-based clustering with those of the deepest wells, we suggest their agreement, although very approximate. Obviously, each deepest well on $FEL_{RMSD}^{R_{g}}$ is formed by the most similar KID conformations of close size then the most populated cluster contained the more heterogeneous conformations.

## 3. Discussion

In the present study, we focused on the kinase insert domain of RTK KIT, the key recruitment region for a host of downstream signalling proteins—enzymes and adapter/scaffolding proteins [30]. Using the MD simulation-based study, we showed that the KIT kinase insertion domain is intrinsically disordered, and this property is disconnected from the KID context, as a cleaved polypeptide or as a domain of the receptor. Intrinsic disorder is a central feature in the function of numerous proteins, enabling them to serve as molecular switches and hubs in biological networks [31,32]. The intrinsically disordered regions (IDRs) of these proteins are increasingly recognised for their prevalence and their critical roles in regulatory intermolecular interactions. It has been estimated that IDRs in the human proteome contain ~132,000 binding motifs [33]. Disordered proteins are believed to account for a large fraction of all cellular proteins, playing roles in cell-cycle control, signal transduction, transcriptional and translational regulation, and large macromolecular complexes [34].

Principally, the RTK KIT contains three regions, JMR, KID and C-terminal, which retain the phosphotyrosine residues regulating signalling in multiple ways. The activation loop (A-loop) of KIT holds also two tyrosine residues, Y812 and Y823, the functions of which are still under discussion [35,36]; however, it was demonstrated that the mutation of Y823 causes aberrant downstream signalling and a possible binding of proteins by the phosphorylated Y823 was suggested [36]. Consequently, these four phosphotyrosine-containing regions of KIT are the principal regulatory subunits, maintaining the spatiotemporal control of KIT signalling.

In KIT, these regions are mainly composed of polar and charged residues, reaching up to 58% in KID and C-terminal, 46% in JMR and 43% in A-loop, while the portion of hydrophobic residues is reduced to 34% in JMR and KID, and 36% in C-terminal, while in A-loop their amount reach up to 47%. High content of polar and charged residues and a low population of hydrophobic residues is an archetypal signature of the intrinsically disordered proteins [21,37]. The lower portion of charged and polar residues in JMR and A-loop compared to KID and C-term, is reflected in the enhanced stability of their 2D and 3D structure in each receptor’s state, inactive or active, as shown in crystallographic structures [38,39]. At the same time, A-loop and JMR, compared to the other KIT regions, show the largest difference in their position and structure in the inactive and active states (RMSD of 17 and 10 Å for A-loop and 5 and 2 Å for JMR), suggesting their intrinsic disorder. While KID is longer in sequence and has a higher fraction of disordered residues compared to the other phosphotyrosine-containing KIT regions, its intrinsic disorder is more abundant.

The intrinsic disorder of KID makes it difficult to study by experimental techniques [40], and the 3D numerical model is the unique description of KID from KIT [11]. By taking these 3D data as the initial structure for the study of its evolution over time, we obtained an accurate and detailed atomistic physical model which presents the intrinsically disordered kinase insertion domain of KIT as a set of heterogeneous conformations. The ensembles of conformations generated for each studied KID entity were characterised by using statistical descriptors which are tightly related with either the geometrical (structural) features or the physical properties.

Our results show that accurate physical models of flexible biomolecular systems, such as proteins consisting of several intrinsically disordered regions, are well founded and can serve to pave the way to establishing the relationship between conformational flexibility and biological function.

The KID folding displays a collection of highly variable helices altered in length and type (α- or 3_10_-helices). Except for H1, a unique helix constantly folded as an α-helix and varying only in its length, the other KID helices are transient, converting between the α- and 3_10_-helices or a helix and non-regular secondary structures—turns, bends, and coils. Apparently, the helical fold is a proper feature of KID, disconnected from KID status (as a cleaved species or a domain fused to KIT) and of a context of MD simulations (unconstrained or restrained distance F689-D768). Nevertheless, a quantity of such folding depends partially not only on the restriction applied to KID extremities, but also on the kind of such restriction. First, a greater degree of the 2D folding was detected in the cleaved KID (KID^C^ and KID^CR^), evidenced by the arrangement of transient helices H4 and H6, not observing in KID fused to KIT (KID^D^). Moreover, comparing 2D structures of the cleaved KID, we observed that H2 and H3 helices are more extended in KID simulated with restrained N- and C-ends (KID^CR^).

The conserved α-helical architecture of the H1 helix, reported for KID embedded to KIT [11], is also observed in each studied KID entity, independently from its context—KID fused to KIT or cleaved polypeptide. Such structural consistency, together with conservation of its spatial position, suggests that this helix, immediately adjacent to the kinase domain, is the inner ordered motif of KID, which is critical for KIT function. Indeed, despite the changes in the helix length along the same trajectory or in different trajectories of distinct KID entities, H1, like a drop of glue, which attaches the KID structural fragments around them by manifold contacts and stabilises a ‘globule-like’ shape of the intrinsically disordered KID. The ‘organising role’ in stabilising the KID structure was previously attributed to tyrosine Y747 located on the helix H4 [11]. We suggest that the Y747 and H1-helix functions are complementary and can be mutually dependent.

The reduced portion of folded secondary structure in KID fused to KIT (KID^D^) suggests that its folding is controlled by the kinase domain. We postulate that the KID folding is principally sequence dependent, but partly allosterically regulated by the kinase domain of KIT. Allostery within or mediated by intrinsically disordered proteins ensures robust and efficient signal integration through mechanisms that would be extremely unfavourable or even impossible for ordered globular protein upon its interaction with the partners [41,42].

The other inherent structural feature of KID is related to flexible linkers connecting these helices, which favour many relative orientations of helices, leading to a vast set of highly heterogeneous conformations. Apparently, the conformational flexibility may be more important than the specific secondary structures before binding. The length of KID linkers is variable and strongly depends on the order of helical folding. Our study revealed that despite great flexibility, once in either a cleaved state or fused to KIT, KID acquires a compact globule-like shape, with a helical fold fraction ranging from the poor (in KID^D^) to rich (KID^C^). A helical structure is, in general, not stable by itself—short helices taken out from stable globular proteins are found to be unstable when isolated [43]. Additional stabilising interactions of the helical structure must, therefore, be present in globular proteins.

In KID, the bulky hydrophobic residues of the weakly fluctuating regions localised on ordered structures (H1 and H5) or random coil (linker between H3 and H4/H5), interact strongly between themselves, forming a ternary hydrophobic core which stabilises the entire tertiary structure of the helically folded KID. The stabilisation of the N- and C-termini of KID is achieved by switchable H-bonds formed by terminal residues in various combinations in different KID entities.

The intrinsically disordered KID contains three known phosphorylation sites (residues Y703, Y721 and Y730) that control KIT signalling. Tyrosine Y703 is located on the αH1-helix, the most conserved structure compared to the rest of KID, although its length varies greatly in each KID entity. The other phosphorylation sites are located on fully transient structures. These observations raised the possibility that, when targeting signalling proteins, KID could display a rich landscape of overlooked folding/dynamic properties. We suggest that (i) transient helices are structures preconfigured to specifically localise signalling proteins, which are selective for alternative phosphotyrosine sites of KID, and may facilitate phosphorylation-mediated regulation of signalling cascade, and (ii) KID has evolved an unusual structural flexibility to stabilise long-lived folding intermediates and, thus, maximise their ‘recognisability’ to signalling proteins. The specificity of intermolecular interactions of KID with signalling proteins is apparently determined by sequence- and structure-based selectivity, which are the two determining factors in ‘molecular recognition’. Furthermore, the transient helices provide inherent stability near phosphorylation sites in KID, consistent with the finding of short and transient helical structures near the phosphorylation sites of the (unphosphorylated) disordered region of the Src homology 2 protein domains (SH2) of adaptor or signalling molecules—Grb2, PI3K, PLCγ [5].

Several questions/issues naturally arise regarding fundamental aspects of the folding/unfolding process in KID, such as the following: (i) explicitly characterising the free energy landscape of folding, which, as we found, is coupled to the KID status and naturally to its binding properties; (ii) to choose topographic characteristics as a criterion for referencing a cleaved KID as the appropriate species for the study of post-transduction process. Furthermore, it is of significant interest to explore to what extent the funnel landscape perspective also applies to intrinsically disordered proteins.

The first principles describing the protein properties—the size (the radius of gyration, Rg), the RMSD (the measure of the average distance between the conformations), the amount of the organised secondary structure (the helical folding order parameter, $Hel_{fp}$) and the solvent accessible surface area (SASA)—were used as reaction coordinates (collective variables) for the evaluation of the relative free energy (ΔG) of KID. Similar measurements have been used to compute interesting protein physics [44,45,46]. Furthermore, such first-principles-based descriptions of the free energy landscape of a disordered protein could be a part of machine learning algorithms that attempt to find energy functions that will finely represent the protein landscape.

The native structure of a protein, as stated in free energy theory [47], shows the lowest free energy in the large conformational space. In this case, the free energy landscape of protein folding is funnel shaped and oriented towards a single well (basin) corresponding to a native structure. The ‘trapping’ conformations observed during the folding process show a shallower well depth than the overall bias towards the native structure, which ensures that the native structure is both thermodynamically favourable and kinetically accessible. This is valid in the case of a well-ordered protein, whereas the IDP we suggest has several ‘quasi-native’ states. Despite some conventionality in using RMSD as a reaction coordinate, the free energy landscape defined on Rg and RMSD showed a series of well-resolved wells, which can be interpreted as ‘quasi-native’ states of intrinsically disordered KID. Landscapes defined on the basis of structurally or physically related measures (SASA, Rg or order of helicity), used as reaction coordinates, displayed a single deep and enlarged well.

We show that the cleaved unconstrained KID explores multiple conformational substates, which are represented by conformations ranging from compact to lose, regardless of their helical content, and showed that the solvent-accessible surface area is highly dependent on the radius of gyration. KID with ends naturally restricted in KID^D^ or mimicking a native restriction in KID^CR^ exhibit a detectable reduction in helical structure and reduced conformational variability (diminished number of substates). Most of the conformations forming the deepest well on the FELs of KID^CR^ are characterised by high compactness and reduced solvent accessible surface area, while in KID^D^, this well is completed by semi-compact conformations that have a larger solvent-accessible surface area than in KID^CR^ and comparable with those in KID^C^. In general, KID has been found to occupy numerous conformational substates in the weakly funnelled free energy landscape, which distinguishes it from the highly directed, time-dependent free energy landscape of regularly folded globular proteins [48,49,50].

The free energy landscapes constructed from first principles describe KID conformations through a collection of minima (so-called basins) and provide a direct ‘polymetric’ evaluation of conformational ensemble each KID species and a comparison between the conformational ensembles of different KID entities. This description is much more accurate than grouping conformations using ensemble-based clustering and allows direct comparison between proteins based on structural metrics or physically related parameters.

By comparing the respective species, we found that the profiles of the free energy landscapes of KID^D^ and KID^CR^ are approximately identical, with only one deep well, while in the unconstrained cleaved KID the free energy landscape is highly disrupted, which leads to slightly different wells with clearly pronounced minima. This indicates that the conformational ensemble of the cleaved KID simulated using restricted N- and C-termini (KID^CR^), better reproduces the KID from the KIT.

Therefore, for studying post-transduction events of KIT, KID^CR^ is the most suitable entity. We suggest that the N- and C-termini of the cleaved KID can be stabilised by a linker, which is a chemical or polypeptide fragment 10 Å in length. Such a cyclic polypeptide would be a *bona fide* generic molecule mimicking the native KID for both computational and empirical studies of KID phosphorylation and recruitment of a multitude of signal transducers to its docking sites. Nevertheless, it seems to us that the use of the cleaved KID cannot be universal. Even for the study of individual downstream signalling pathways mediated, for example, by KIT possessing ‘oncogenic driver’ mutations in the tyrosine kinase domain, a full-length cytoplasmic domain would be required.

## 4. Materials and Methods

### 4.1. 3D Models

The 3D model of the full-length cytoplasmic domain of KIT was taken from [11]. The coordinates of kinase insert domain (KID, sequence F689–D768 aas) were extracted and used as a model of the cleaved KID.

### 4.2. Molecular Dynamics Simulation

#### 4.2.1. Preparation of the Systems

For MD simulation, models of KIT and KID (inactive unphosphorylated state) were prepared with the LEAP module of Assisted Model Building with Energy Refinement (AMBER) [51] using the *ff99SB* all-atom force field parameter set: (i) hydrogen atoms were added, (ii) covalent bond orders were assigned, (iii) protonation states of amino acids were assigned based on their solution for pK values at neutral pH, (iv) histidine residues were considered neutral and protonated on ε-nitrogen atoms, and (v) Na^+^ counter-ions were added to neutralise the protein charge.

Each protein was solvated with explicit TIP3P water molecules in a periodic rectangular box with at least 12 Å distance between the proteins and the boundary of the water box. The total number of atoms in the systems (protein, water molecules and counter ion) was 69,089 and 19,537 for the KIT and KID, respectively.

#### 4.2.2. Setup of the Systems

The setup of the systems was performed with the Simulated Annealing with NMR-Derived Energy Restraints (SANDER) module [52] of AMBER. First, each system was minimised successively using the steepest descent and conjugate gradient algorithms as follows: (i) 10,000 minimisation steps where the water molecules have fixed protein atoms, (ii) 10,000 minimisation steps where the protein backbone is fixed to allow protein side chains to relax, and (iii) 10,000 minimisation steps without any constraint on the system. After relaxation, each system was gradually heated from 10 to 310 K at constant volume using the Berendsen thermostat [53] while restraining the solute Cα atoms by 10 kcal/mol/Å^2^. Thereafter, the system was equilibrated for 100 ps at constant volume (NVT) and for a further 100 ps at constant pressure (NPT) maintained by the Monte Carlo method [54]. Final system equilibration was achieved by a 100 ps NPT run to assure that the water box of the simulated system had reached the appropriate density. The electrostatic interactions were calculated using the PME (particle mesh Ewald summation) [55] with a cut-off of 10.0 Å for long-range interactions. The initial velocities were reassigned according to the Maxwell–Boltzmann distribution.

#### 4.2.3. Production of the MD Trajectories

All trajectories were produced using the AMBER ff99SB force field with the PMEMD module of AMBER 16 and AMBER 18 [51] (GPU-accelerated versions) running on a local hybrid server (Ubuntu, LTS 14.04, 252 GB RAM, 2× CPU Intel Xeon E5-2680 and Nvidia GTX 780ti) and the supercomputer JEAN ZAY at IDRIS.

The multiple extended trajectories were generated for each equilibrated system: two 2-µs trajectories for KIT with KID, four 1.8-µs replicas for cleaved KID (KID^C^) and two 1.8-µs replicas for cleaved KID with the restrained distance (10 Å) between the Cα-atoms of terminal residues, F689 and D768 (KID^CR^).

A time step of 2 fs was used to integrate the equations of motion based on the Leap-Frog method. Coordinate files were recorded every 1 ps. Neighbour searching was performed by the Verlet algorithm [56]. The Particle Mesh Ewald (PME) method, with a cut-off of 10 Å, was used to treat long-range electrostatic interactions at every time step. The van der Waals interactions were modelled using a 6–12 Lennard–Jones potential. The initial velocities were reassigned according to the Maxwell–Boltzmann distribution. Coordinates were recorded every 1 ps.

#### 4.2.4. Data Analysis

Unless otherwise stated, all recorded MD trajectories, individual and merged, were analysed (RMSFs, RMSDs, DSSP, clustering) with the standard routines of the CPPTRAJ 4.15.0 program [57] of AMBER 20 Suite. All analysis was performed on the MD conformations (every 10 ps) by considering either all simulations or the production part of the simulation, which was generated after the removal of non-well-equilibrated conformations (0–70 ns), as was shown by the RMSDs, or on residues with a fluctuation of less than 4 Å, as shown by the RMSFs, and after least-square fitting [58,59] of the MD conformations for a region of interest, thus removing rigid-body motion from the analysis.

(1)The RMSD and RMSF values were calculated for the Cα-atoms using the initial model (at *t* = 0 ns) as a reference.(2)Secondary structural propensities for all residues were calculated using the Define Secondary Structure of Proteins (DSSP) method [60]. The secondary structure types were assigned for residues based on backbone -NH and -CO atom positions. Secondary structures were assigned every 10 and 20 ps for the individual and concatenated trajectories, respectively.(3)Clustering analysis was performed on the productive simulation time of each MD trajectory using an ensemble-based approach [24]. The analysis was performed every 100 ps. The algorithm extracts representative MD conformations from a trajectory by clustering the recorded snapshots according to their Cα-atom RMSDs. The procedure for each trajectory can be described as follows: (i) a reference structure is randomly chosen in the MD conformational ensemble, and all conformations within an arbitrary cut-off r are removed from the ensemble; this step is repeated until no conformation remains in the ensemble, providing a set of reference structures at a distance of at least r; (ii) the MD con-formations are grouped into n reference clusters based on their RMSDs from each reference structure. The cut-off was varied from 3 to 5 Å.(4)H-bonds between heavy atoms (*N*, *O*, and *S*) as potential donors/acceptors were calculated with the following geometric criteria: donor/acceptor distance cut-off was set to 3.6 Å, and the bond angle cut-off was set to 120°. Hydrophobic contacts were considered for all hydrophobic residues with side chains within a 4 Å of each other.(5)Contact search was performed with Cpptraj. Contact present if Cα–Cα distance <10 Å. Map normalised over the number of frames such as the value represents the contact frequency in the considered trajectory.(6)The radius of gyration (Rg) was calculated from the atomic coordinates using the formula (Equation (1)) from [61]:(1)$$R_{g}=\sqrt{\frac{\sum_{i=1}^{N}m_{i}r_{i}^{2}}{\sum_{i=1}^{N}m_{i}}}$$
where $m_{i}$ is the mass of the atom i, $r_{i}$ is the distance of atom i from the center of mass of the protein.(7)The relative Gibbs free energy of the canonical ensemble was computed as a function of two reaction coordinates (Equation (2)) [62]:(2)$$\Delta G_{\left(R1,R2\right)}=-k_{B}T\ln\frac{P_{\left(R1,\;R2\right)}}{P_{max}}$$
where $k_{B}$ represents the Boltzmann constant, *T* is the temperature. $P_{\left(R1,\;R2\right)}$ denotes the probability of states along the two reaction coordinates, which is calculated using a k-nearest neighbour scheme and $P_{max}$ denotes the maximum probability. The 3-dimensional representations of the free energy surface were plotted using Matlab (© 1994-2021 The MathWorks, Inc., Natick, MA, USA).

#### 4.2.5. Advanced Data Analysis

Secondary structure for each KID residue of each replica (DSSP) was classified by an 8-letter code [63] and used for calculation of an estimated transition probability matrix from one folding state to another. A suitable distance (Fisher-Rao distance) between these families was calculated for all the pairs of replicas. Multi-Dimensional Scaling (MDS) was performed to get an ‘as isometric as possible’ embedding of the data in 2D (i.e., a representation by placing points on a plane while preserving the calculated inter-distances as well as possible).

#### 4.2.6. Visualisation and Figure Preparation

Visual inspection of the conformations and figure preparation were performed with PyMOL (https://pymol.org/2/, accessed on 14 September 2020). The VMD 1.9.3 program (accessed on 21-12-2020) [64] was used to prepare the protein MD animations. To visualise the motions along the principal components, the Normal Mode Wizard (NMWiz) plugin [65], which is distributed with the VMD program, was used.

## Figures and Tables

**Figure 2 ijms-22-07375-f002:**
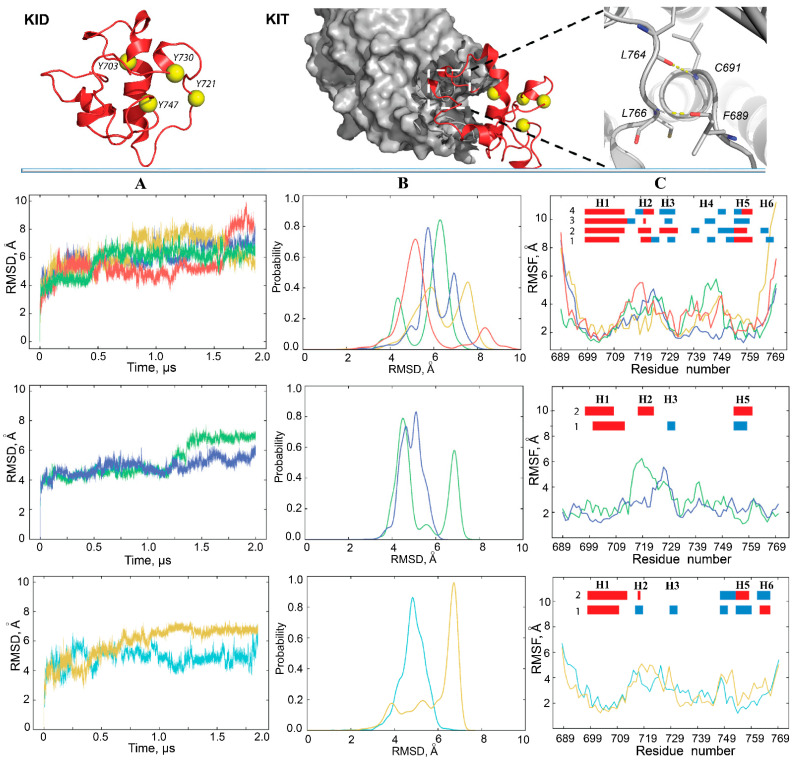
Conventional MD simulations of KID. (Top panel) Structural models of the cleaved KID (KID^C^) (left) and KID fused to KIT (KID^D^) (middle) in which the N- and C-ends are stabilised by pairs of H-bonds (right). Proteins are displayed as ribbons (KID), surface-filled model (kinase domain), yellow balls (the tyrosine residues) and sticks (residues formed H-bonds). Columns (**A**–**C**) show the statistical descriptors of KID^C^ (top), KID^D^ (middle) and KID^CR^ (bottom) computed on the all Cα-atoms of KID for MD conformations of each trajectory after fitting on initial conformation. (**A**) RMSDs, (**B**) probability distributions of the RMSDs, and (**C**) RMSFs computed on the Cα atoms after fitting on initial conformation. In the insert, the folded secondary structures, αH- (red) and 3_10_-helices (blue), labelled as H1–H6, were assigned for an average conformation of each MD trajectory. In (**A**–**C**), MD replicas are distinguished by colour: 1–4 of KID^C^ are in green, yellow, blue, and red; 1–2 of KID^D^ are in green and blue; 1–2 of KID^CR^ are in blue light and yellow respectively.

**Figure 3 ijms-22-07375-f003:**
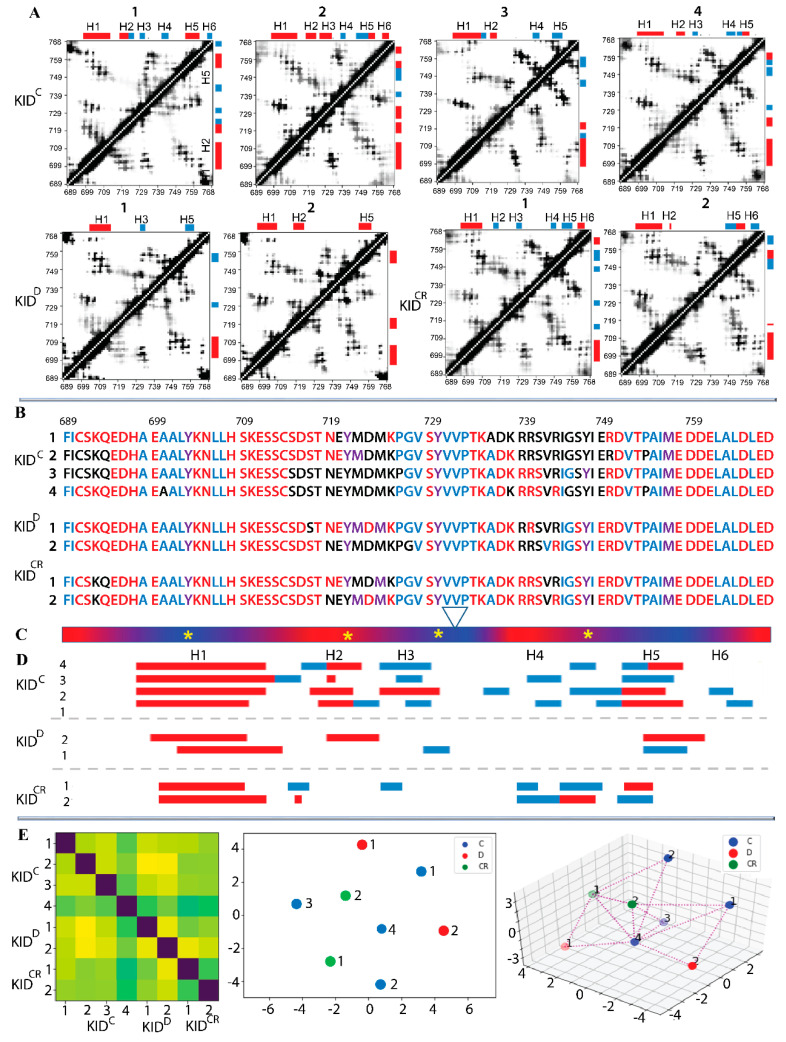
Estimation of intramolecular contacts in KID. (**A**) Dynamic contact maps of pairwise distances Cα−Cα < 10 Å computed for each MD trajectory. Black–white gradient shows a frequency (from 1 to 0) of contact during a trajectory. Secondary structure bars are shown at the top and on the left of the contact map. (**B**) Residues participating in steady contacts (in colour) were identified from (**A**) and analysed visually (PyMOL). Residues from the ‘pseudo-rigid’ segments E699–S709, P754–L764 and V731–P733 were used as an origin for the computing of contacts to all residues of KID. Only the contacts with an occurrence greater than 80% were considered. Polar, hydrophobic, and amphiphilic residues are denoted in red, blue and violet respectively. (**C**) Red–blue gradient shows the RMSF values, from large (>4 Å, in red) to small (<2 Å, in blue). The position of tyrosine residues is shown by the yellow asterisks. (**A**,**D**) The αH- (red) and 3_10_-helices (blue), were assigned by DSSP on average conformation from each MD trajectory and labelled from H1 to H6. (**E**) The per-residue modelling by finite-state Markov models of the secondary structure dynamics of KID. Transition probability matrix from one folding state to another (from one letter to another) was obtained on data encoding the secondary structure for each residue of each replica. (Left) The Fisher-Rao matrix (8 × 8 of size), where the first 4 replicas are the cleaved KID^C^ (group C), the next two the KID^D^ (group D), and the last two the cleaved restricted KID^CR^ (group CR). (Right) Multi-Dimensional Scaling (MDS) represents an ‘as isometric as possible’ embedding of the data in 2D and 3D (i.e., a representation by placing points on a plane and in Cartesian coordinates while preserving the calculated inter-distances as well as possible).

**Figure 4 ijms-22-07375-f004:**
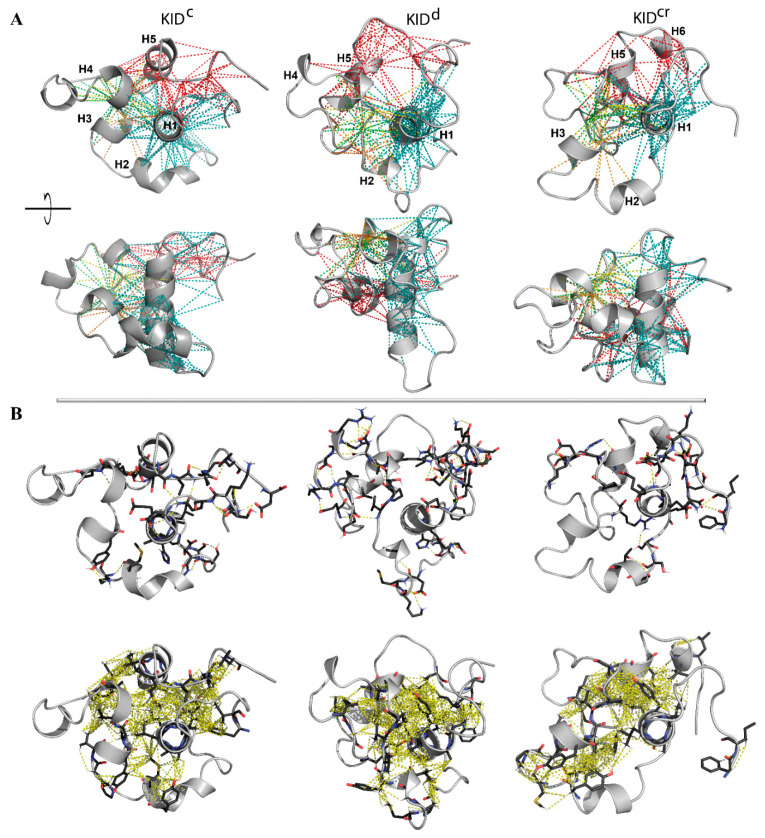
Non-covalent interactions maintaining the inherent (intrinsic) 3D structure of KID. (**A**) The intramolecular contacts, H-bonds and hydrophobic interactions (dashed lines), formed by residues E699–709 (H1) (in teal), P754-L764 (H5) (in red) and V731/V732/P733 (orange/green/yellow) with all the residues of KID at least in one trajectory, are superimposed on a randomly chosen KID conformation shown in two orthogonal projections. Protein is shown in grey ribbons, helices H1–H6 are labelled for KID^C^. (**B**) H-bonds (D-H⋯A ≤ 3.6 Å, where D (D = O/N/S) is a donor atom and A (A = O/N/S) is an acceptor atom) (top) and van der Waals contacts (C-H⋯A = O/N/S ≤ 4 Å) between all side chain atoms (bottom) are shown on a randomly chosen conformation. (**A**,**B**) The interactions stabilising the regular structures (helices) are not considered.

**Figure 5 ijms-22-07375-f005:**
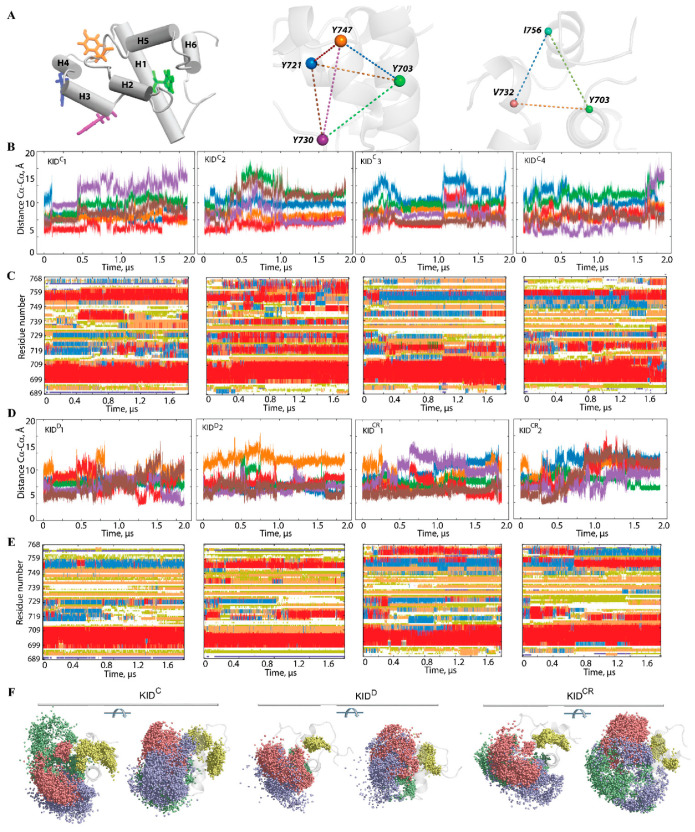
The inter-residue geometry of tyrosines in the isolated unconstrained KID and its relationship with folding. (**A**) KID structure with helices shown as solid cylinders and tyrosine residues as sticks (left), tetrahedron delimited on the Cα-atoms of tyrosine residues (middle) and triangle designed on the most stable among the MD simulation residues (with the smallest RMSF values) (right). (**B**,**D**) Distances between each pair of tyrosine residues in each MD trajectory, coloured as the edges of the tetrahedron. (**C**,**E**) The time-related evolution of the secondary structures of each residue as assigned by DSSP with the type-coded secondary structure bar. (**D**) Distances between the most ‘stable’ (minimal RMSF values) residues Y703, V732 and I756 over each cMD trajectory, coloured as the edges of the triangle in (**A**). (**F**) The spatial distribution of hydroxyl groups presented by the oxygen atoms of the tyrosine residues in KID is shown in two orthogonal projections with the oxygen atoms of Y703, Y721, Y730 and Y747, respectively coloured in yellow, blue, red and green.

**Figure 6 ijms-22-07375-f006:**
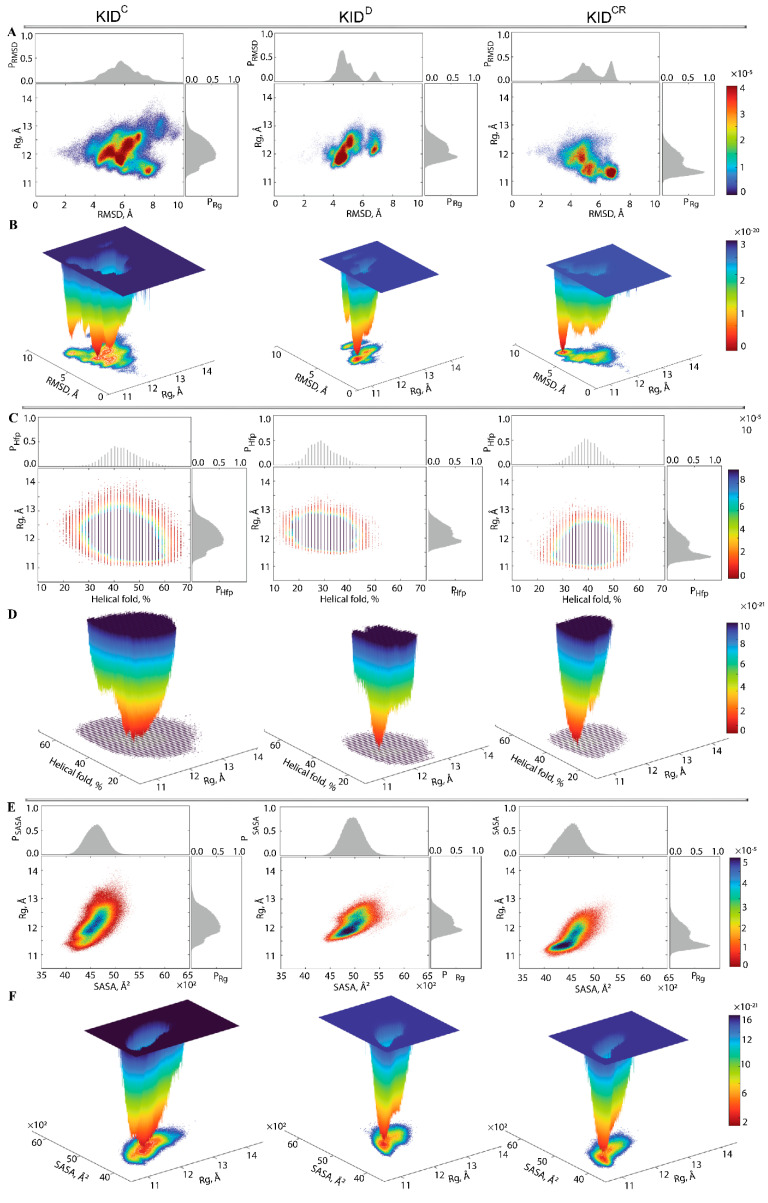
Free energy landscape (FEL) of KID as a function of the reaction coordinates. Each FEL generated on the MD conformations of different KID entities (KID^C^, KID^D^ and KID^CR^) is displayed for the conformational ensemble sampled on the merged replicas. Calculations of the radii of gyration are performed on the Cα-atoms. The 2-dimensional representation of FEL of the KID conformational ensembles plotted as a function of Rg (in Å) versus (**A**) RMSD (in Å); (**C**) $H_{fp}$, which describes the fraction of helices (as calculated by the DSSP) (in %); (**E**) SASA, which describes the solvent-accessible surface area (as calculated with Cpptraj) (in Å^2^). Probability distribution of each reaction coordinate is shown at the top and right, respectively. The red colour represents high occurrence, yellow and green low, and blue represents the lowest occurrences. The 3-dimensional representation of the relative Gibbs free energy as a function of Rg and (**B**) RMSD; (**D**) $H_{fp}$; (**F**) SASA. The blue colour represents the high energy state, green and yellow low and blue represents the lowest stable state. The free energy surface was plotted using Matlab.

**Figure 7 ijms-22-07375-f007:**
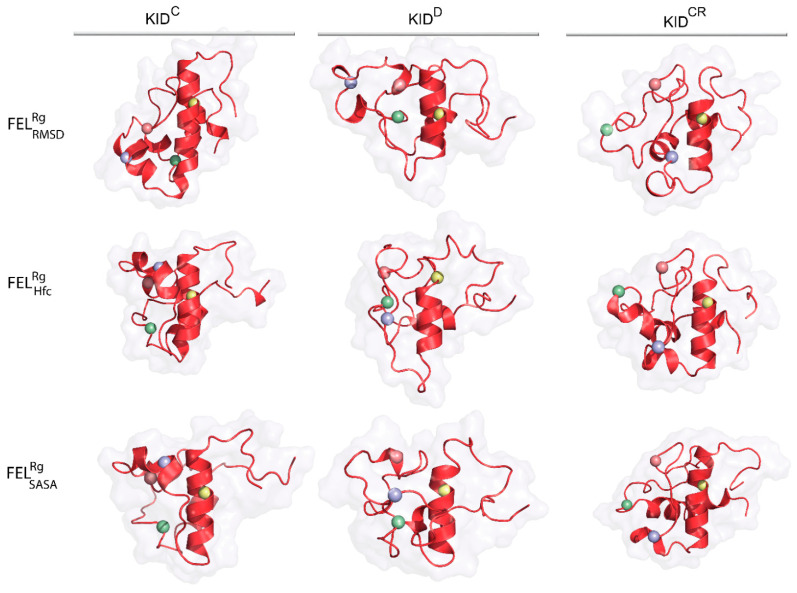
Representative conformations of the deepest well on the free energy landscape (FEL) of KID. KID displayed as red ribbons contoured with a surface-filled model. Position of the tyrosine residues (Cα-atoms) is shown as balls coloured in yellow, blue, red and green for Y703, Y721, Y730 and Y747, respectively.

## Data Availability

The numerical model simulations upon which this study is based are too large to archive or to transfer. Instead, we provide all the information needed to replicate the simulations. The models’ coordinates are available from L. Tchertanov at ENS Paris-Saclay.

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
