# Peer review of "Folding and Intrinsic Disorder of the Receptor Tyrosine Kinase KIT Insert Domain Seen by Conventional Molecular Dynamics Simulations"

_ijms, 2021, doi:10.3390/ijms22147375_

Round 1

Reviewer 1 Report

The RTK KIT is extremely important for the transmission of cellular signals, it’s malfunction may cause severe deseases. The MS by Julie Ledoux and colleagues devoted to the MD modeling of it’s key region, kinase insert domain, KID, is of special interest as this domain has no well-defined structure. The MD-based structural model of KID was recently proposed by the group with the same corresponding author L.Tchertanov (Scientific Reports,2020;10(1):5401). The current MS may be considered as a continuation of the previous studies. Here the authors consider three MD approaches that can be used to model the behavior of KID and study, for example, the post-transduction effects and stand for one, which is the modeling of the cleaved domain with the distance restrained terminal residues.

I have no major objections to the work. The aims, results and discussion are written clear enough though might be a bit more condensed.

Some minor comments and questions are listed below:

[1] When the whole KIT structure was taken into account, were there any attempts to apply some restrictions to mimic the limited mobility of the transmembrane domain and/or SCF linkage between the neighbors?

[2] Why a Langevin piston (and not say Parrinello-Rahman) was used for pressure coupling in the NPT ensemble?

[3] How was “a k-nearest neighbour scheme” implemented? (line 802-803 p.24) Could some more details be provided.

[4] Is a computer RAM value correct: 252 GB RAM? (line 747 p. 22)

Usually GBs are the power of 2.

Style:

[a] line 20: I would avoid ‘basins’ in the abstract. Why not commonly used ‘wells’.

[b] An ‘average structure’ is more frequently used in literature than ‘mean’.

[c] Some mixture of British and American English through the MS leaves the vivid impression of multi-authorship. Note ‘stabilising’ in line 353 and ‘stabilizing’ in 355.

Misprints:

line 663: ‘qiasi-native states’ → ‘quasi-…’

line 664: ‘lanscape’ → ‘landscape’

line 665: ‘well-reasolved bassins’ → ‘well-resolved basins’

line 666-668: ‘Landscapes defined on structurally or physically related mesures (SASA, ?? or order of helicity) as reaction coordinates displied a single deep and enlarged basin’ - please rephrase. Are ‘mesures’ and ‘displied’ correct?

Fig S1 legend: ‘fist’ →’first’

Author Response

Responses to Reviewer #1:

Comments: The RTK KIT is extremely important for the transmission of cellular signals, it’s malfunction may cause severe deseases. The MS by Julie Ledoux and colleagues devoted to the MD modeling of it’s key region, kinase insert domain, KID, is of special interest as this domain has no well-defined structure. The MD-based structural model of KID was recently proposed by the group with the same corresponding author L.Tchertanov (Scientific Reports,2020;10(1):5401). The current MS may be considered as a continuation of the previous studies. Here the authors consider three MD approaches that can be used to model the behavior of KID and study, for example, the post-transduction effects and stand for one, which is the modeling of the cleaved domain with the distance restrained terminal residues.

I have no major objections to the work. The aims, results and discussion are written clear enough though might be a bit more condensed.

Response: The Authors thank Reviewer 1 for positive comments of the manuscript and critical remarks. Our responses are supplied after each concern.

Some minor comments and questions are listed below:

[1] When the whole KIT structure was taken into account, were there any attempts to apply some restrictions to mimic the limited mobility of the transmembrane domain and/or SCF linkage between the neighbors?

Response: The cytoplasmic domain (CD) of KIT was considered as a fully relaxed model of CD in solution without any restriction. This model is not influenced by CSF, as it represents the inactive form of KIT (a monomer) and not attached to the membrane. To avoid rotational/translational motions of KIT as an entire body, all collected frames were normalised by a fitting on the first conformation (initial, at t=0). Looking ahead, I precise that we are currently working for a search of inter-dependence between different regions (JMR, N- and C-lobes, A-loop, KID and C-terminal) to understand how conformational dynamics orchestrates KIT functions in allosteric regulation of recognition and catalysis. In this case, the CD is attached through JMR to the transmembrane helix inserted into membrane, mimicking the natural environment of KIT.

[2] Why a Langevin piston (and not say Parrinello-Rahman) was used for pressure coupling in the NPT ensemble?

Response: The barostat is a Monte Carlo. Thank you for pointing out this part of the Materials and Methods. The generalized shadow hybrid Monte Carlo method, incorporated in AMBER, allows a rigorous control of a system at constant pressure and constant temperature without a loss of computational efficiency.

[3] How was “a k-nearest neighbour scheme” implemented? (line 802-803 p.24) Could some more details be provided.

Response: In statistics, the k-nearest neighbour algorithm (k-NN) is a non-parametric classification method first developed by Fix and Hodges (1951) and later evaluated by Cover and Hart (1967). The k-NN method is used for classification and regression.

 [4] Is a computer RAM value correct: 252 GB RAM? (line 747 p. 22)

Usually GBs are the power of 2.

Response: The computer RAM value is correct. A high-performance computer cluster having a memory of 252 Gb with a swap of 67 Gb was used.

Style:

[a] line 20: I would avoid ‘basins’ in the abstract. Why not commonly used ‘wells’.

Response: In the abstract and text, ‘basins’ was changed for ‘wells’.

[b] An ‘average structure’ is more frequently used in literature than ‘mean’.

Response: The term ‘mean structure’ was changed for ‘average structure’.

[c] Some mixture of British and American English through the MS leaves the vivid impression of multi-authorship. Note ‘stabilising’ in line 353 and ‘stabilizing’ in 355.

Response: The text was unified for British English.

Misprints:

line 663: ‘qiasi-native states’ → ‘quasi-…’

Response: It was changed for ‘quasi-native states’.

line 664: ‘lanscape’ → ‘landscape’

Response: The keyboard error was corrected.

line 665: ‘well-reasolved bassins’ → ‘well-resolved basins’

Response: The keyboard error was corrected.

line 666-668: ‘Landscapes defined on structurally or physically related mesures (SASA, ?? or order of helicity) as reaction coordinates displied a single deep and enlarged basin’ - please rephrase. Are ‘mesures’ and ‘displied’ correct?

Response: The sentence was changed for: ‘Landscapes, defined on structurally or physically related measures (SASA, ?? or order of helicity) used as reaction coordinates, displayed a single deep and enlarged well.’

Fig S1 legend: ‘fist’ →’first’

Response: The keyboard error was corrected.

Reviewer 2 Report

See attached file

Author Response

Responses to Reviewer #2:

This is an interesting study on the important topic of intrinsically disordered domains (ID) and their role in signaling. It follows up on an earlier proposal by the authors of a 3D model of the KIT receptor tyrosine kinase KID domain. In the present study, the authors compare the energetics and dynamics of this domain in isolation with it embedded within KIT. New insights are presented on the conservation of a core set of contacts, the collective dynamics of the four phospho-tyrosine residues and methodology for construction of ID landscapes. The methods section is well-described.

Response:  The Authors thank Reviewer 2 for very positive comments and critical reading of our manuscript. Our responses are supplied after each concern.

However, I have problems with the Discussion and the emphasis given by the authors to one result – the fact that KID dynamics when constrained by linkers between the termini are more to the embedded domain than the isolated one. First, this should not be a great surprise.

Response: You are right, this finding is not surprising, but it was neither shown nor proven.

Second, the results presented to support this claim argue that the constraints only partially restore the native state, but the authors gloss over this point. Some discussion of what factors might explain this difference would be desirable.

Response: We found the KID folding depends on (i) the restriction applied to KID extremities and (ii) the kind of such restriction. These are the observations. Next, we postulate that the KID folding is principally sequence-dependent but partly allosterically regulated by the kinase domain of KIT. This is a hypothesis, and the reasoning on this matter seems, to me, not very timely. We can suggest the allosteric regulation of KID by kinase domain, but to assert this hypothesis as a fact and to discuss it, strong evidences are required, such as coupled movements, communications, etc. Looking ahead, I precise that we are currently working on inter-dependences between different regions of KIT (JMR, N- and C-lobes, A-loop, KID and C-terminal) to understand how conformational dynamics orchestrates KIT functions in allosteric regulation of recognition and catalysis.

To me, the most consequential result is the discovery of a conserved set of core contacts organized around a central helix H1 (Figure 4). A similar design is obtained in other signal domains, the armadillo domain for example (Pandini et al., 2015). A major rewrite of the Discussion is needed to make it more concise and achieve a better balance between the simulation results before publication.

Response: Thanks for your suggestion. The following text was inserted into the text (lines 600-610):

“The conserved α-helical architecture of the H1 helix, reported for KID embedded to KIT (ref), is also observed in each studied KID entity, independently from its context - KID fused to KIT or cleaved polypeptide. Such structural consistency, together with conservation of its spatial position, suggests that this helix, immediately adjacent to the kinase domain, is the inner ordered motif of KID, which is critical for KIT function. Indeed, despite the changes in the helix length along the same trajectory or in different trajectories of distinct KID entities, H1, like a drop of glue, which attaches the KID structural fragments around them by manifold contacts and stabilises a ‘globule-like’ shape of the intrinsically disordered KID. The ‘organising role’ in stabilising the KID structure was previously attributed to tyrosine Y747 located on the helix H4 (ref). We suggest that the Y747 and H1-helix functions are complementary and can be mutually dependent. “

Minor concerns:

  • The title is wordy, and the nomograms mystify. How about “Folding and intrinsic disorder of the KIT receptor tyrosine kinase insert domain seen by molecular dynamics simulations” or similar.

Response: The abbreviation in the title was decoded. Thank you for the proposed title.

  • Is there a reason when four replicates were performed for the cleaved KID, but only duplicates for the embedded and constrained KIDs?

Response: An expected higher flexibility of the cleaved KID with respect to the embedded and constrained KIDs required a more exhaustive sampling.

  • Figure 8 and the associated text do not add anything. They should be deleted for conciseness and clarity. The text in the abstract that refers to this section could be amended to simply state “We suggest a cyclic, generic KID would be best-suited for future studies of KID post-transduction effects” or similar.

Response: We do not agree with the Referee's suggestion to delete the Figure 8 that shows the structure of the cyclic generic KID, a final product designed after the analysis reported in the manuscript and therefore, is essential to complete the obtained results.  The figure 8 with diminished background around the image was re-inserted.

  • The term “frustrated” in “frustrated contact maps” (Figure 6) should be defined because energy frustration is a specific concept widely used in protein folding studies(Parra et al., 2016).

Response: The most appropriated term ‘dynamic contact map’ was used.

  • The citation to the Fisher-Rao distance (ref.66) seems incomplete. I could not retrieve this work. Please provide a complete, retrievable citation or explain this metric in greater detail in Methods.

Response: The citation was completed.

  • Figure 6. The three coordinate systems used to map the conformational landscape need to be compared more rigorously. In particular, the constrained and cleaved KID landscapes are similar in the RG-RMSD plot but not in the other two where the constrained KID is comparable to the embedded KID. Discuss and explain.

Response: The three used reaction coordinates, RMSD, Rg and SASA, are metrics describing very different quantities - the average distance between the conformations (a metric without any physical significance), the amount of organised secondary structures and the solvent accessible surface area (tow metrics physically significant but not dependent), which are not comparable between them. Only the results for different entities obtained on the same reaction coordinates may be compared. In the text we paid attention to such comparison of KID entities on the qualitative level.

The mentioned by Referee 2 difference of the RMSD-Rg plot of KIDD and KIDCR from that of KIDC arises from the difference of their RMSD profiles which are bimodal in KIDD and KIDCR, large and unimodal in KIDC.

Round 2

Reviewer 2 Report

See attached file

Author Response

Authors: The Authors thanks the Reviewer 2 for a careful reading of our revised manuscript.

The revised manuscript is improved relative to the initial submission. However,

  • The authors misconstrued my critique regarding the conformational landscapes. To be clear, I do not agree with the conclusion that “By comparing the respective species, we found that the free energy landscapes of 702 KIDD and KIDCR are approximately identical – (lines 702-703). They do show “one deep well”, but that does not make them “approximately identical”. Specifically, in the RG versus RMSD landscapes, the KIDCR is intermediate between the KIDC and KIDD Speculation about allosteric mechanism was neither requested, nor is it warranted.

Authors: Thank you for a clearer explication of your concerns. The needed correction was introduced.

  • It is unfortunate that the authors remain fixated on giving one of the lesser results undue prominence via Figure 8. I would alter my opinion if the MD simulations were in fact reported in full in this study, but the statement offered instead “Conformation was taken at 2 μs of MD simulation (full data will be reported as soon as possible)” does not pass credible muster. This statement should be deleted, or the data reported.

Authors: The Figure 8 has been removed. The text was adapted.

  •  The authors have satisfactorily addressed my other concerns.
